# Acute IgA-Dominant Glomerulonephritis Associated with Syphilis Infection in a Pregnant Teenager: A New Disease Association

**DOI:** 10.3390/jcm8010114

**Published:** 2019-01-18

**Authors:** Alejandra Oralia Orozco Guillén, Ricardo Ivan Velazquez Silva, Bernardo Moguel González, Yubia Amaya Guell, Pamela Garciadiego Fossas, Iris Guadalupe Custodio Gómez, Osvaldo Miranda Araujo, Virgilia Soto Abraham, Giorgina Barbara Piccoli, Magdalena Madero

**Affiliations:** 1Department of Nephrology, National Institute of Perinatology “Isidro Espinoza de los Reyes”, Mexico City 11000, Mexico; 2Department of Nephrology, National Institute of Cardiology “Ignacio Chávez”, Mexico City 14080, Mexico; ricardo.ivan.velaz@gmail.com (R.I.V.S.); bernardomoguel@hotmail.com (B.M.G.); 3Department of Foetal Medicine, National Institute of Perinatology “Isidro Espinoza de los Reyes”, Mexico City 11000, Mexico; dra_amaya@hotmail.com; 4Department of Infectolog, National Institute of Perinatology “Isidro Espinoza de los Reyes”, Mexico City 11000, Mexico; drapamelagf@yahoo.com; 5Department of Gynaecology and Obstetrics, National Institute of Perinatology “Isidro Espinoza de los Reyes”, Mexico City 11000, Mexico; iriscustodio15@gmail.com (I.G.C.G.); drosvaldomiranda@gmail.com (O.M.A.); 6Department of Pathology, National Institute of Cardiology “Ignacio Chávez”, Mexico City 14000, Mexico; vigiliasoto@gmail.com (V.S.A.); madero.magdalena@gmail.com (M.M.); 7Department of Clinical and Biological Sciences University of Torino, 10043 Torino, Italy; 8Department of Nephrology, Centre Hospitalier Le Mans, 72000 Le Mans, France

**Keywords:** chronic kidney disease (CKD), preeclampsia (PE), IgA dominant glomerulonephritis, syphilis related glomerulonephritis

## Abstract

Chronic kidney disease (CKD) is increasingly recognized as a risk factor in pregnancy; the differential diagnosis between CKD and preeclampsia (PE) may be of pivotal importance for pregnancy management and for early treatment of CKD. Acknowledging this connection may be useful also in a wider context, such as in the case reported in this paper, which for the first time describes an association between syphilis infection and IgA-dominant glomerulonephritis. A 16-year-old woman, referred to a general hospital due to a seizure, was found to be unknowingly pregnant. Based on hypertension and nephrotic proteinuria, she was initially diagnosed with PE. Immunological tests, as well as hepatitis and HIV tests showed negative results. However, secondary syphilis was diagnosed. In discordance with the PE diagnosis, urinalysis showed glomerular microhematuria with cellular casts. Proteinuria and hypertension did not remit after delivery, which was made via caesarean section, due to uncontrolled hypertension, at an estimated gestational age of 29 weeks. A male baby, weighing 1.1 kg (6.5 centile) was born. The baby was hospitalized in the neonatal intensive care unit, where he developed subependymal hemorrhage and thrombocytopenia, and neonatal syphilis was diagnosed. The mother underwent a kidney biopsy one week after delivery, leading to the diagnosis of IgA-dominant postinfectious glomerulonephritis. Mother and child were treated with support and antibiotic therapy, and were discharged in good clinical conditions four weeks later. Four months after delivery, the mother was normotensive without therapy, with normal kidney function and without hematuria or proteinuria. In conclusion, this case suggests that IgA-dominant postinfectious glomerulonephritis should be added to the spectrum of syphilis-associated glomerulonephritides, and underlines the need for a careful differential diagnosis with CKD in all cases of presumed PE. While diagnosis relies on kidney biopsy, urinary sediment, a simple and inexpensive test, can be the first step in distinguishing PE from other nephropathies.

## 1. Introduction

Pregnancy is a valuable, and often not sufficiently exploited, occasion for diagnosing chronic kidney disease, which may manifest itself for the first time in different forms, from pregnancy-induced hypertension to pregnancy-induced proteinuria or the complete picture of preeclampsia (PE), which encompasses both, up to severe pictures such as eclampsia or HELLP, the acronym for haemolysis, low platelets, elevated liver enzymes [1,2,3]. 

Virtually all chronic kidney diseases, including those characterized only by a reduction of the kidney parenchyma, are associated with an increased risk of PE [4,5,6,7,8,9,10,11]. Interestingly, these forms of “superimposed” PE are associated with a peculiar pattern of angiogenic-antiangiogenic biomarkers, which differs from the more common forms of PE, not related to pre-existing kidney disease [12,13,14,15].

Furthermore, while any clinically latent kidney disease may manifest itself as superimposed PE, the clinical features of glomerular diseases can mimic those of PE, making it is difficult to arrive at a differential diagnosis based solely on the presence of hypertension and proteinuria. This is especially true in cases where the patient’s clinical history is unknown. In such cases, the use of utero-placental Doppler and the analysis of angiogenic-antiangiogenic biomarkers can facilitate the diagnosis. However, these markers are not always available, and placental Doppler seems to be more sensitive in early phases of the condition [12,13,14,15,16,17]. In such a context, especially in settings where resources are limited, other tests, including urinary sediment, should be employed as the basis of the differential diagnosis and to reveal the presence of potentially treatable kidney diseases that would otherwise escape early diagnosis. The importance of timely treatment is obvious, but its relevance is amplified when low availability of renal replacement therapy makes prevention of chronic kidney disease a fundamental goal [18,19].

The case described in this report serves to highlight two mutually linked aspects: the importance of a differential diagnosis between preeclampsia and chronic kidney disease and the role of pregnancy in diagnosing kidney diseases. This case has allowed us to describe a new disease association between syphilitic infection and IgA-dominant glomerulonephritis, and it underlines how examination of kidney ailments in pregnant patients can increase our knowledge about these diseases. 

## 2. The Case

A Hispanic woman, whose estimated age was 16 (the age is uncertain due to the absence of a birth certificate, which was reported lost by her mother), residing in Mexico City, was found unconscious at home by her older sister, who brought her to the nearest general hospital. At hospitalisation she was found to be pregnant, which she, and her family, were apparently unaware of. At referral, she was hypertensive, and high-grade proteinuria (+++) was present at urinalysis. Immediately after the diagnosis of pregnancy, she was referred to the National Institute of Perinatology (INPER), a specialized center for high-risk pregnancy, with a diagnosis of severe PE.

At referral to INPER, the physical examination revealed a small, thin girl (height 1.52 cm, weight 39 kg, BMI: 16.8 Kg/m^2^), with mild peripheral oedema, and no other particular physical finding, except for hypertension (BP: 160/110 mmHg).

Since the date of conception and the date of her most recent menses were unknown, the gestational age was calculated using ultrasounds, and was initially hypothesized as 26–28 weeks. However, given the relationship between the mother’s body size and that of the offspring, it was considered possible that the gestational age was underestimated, and, indeed, at delivery, Ballard score estimated the gestational age at 29 weeks.

The main biochemical tests at referral are reported in Table 1. It is worth noting that all immunological tests performed resulted negative (anti DNA, antinuclear antibodies, LLAC, Anticardiolipin antibodies), while complement fraction C3 was normal (85 mg/dL, normal values 80–160 mg/dL) and C4 was low (7 mg/dL; v 12–70 mg/dL). The patient’s biochemical markers indicated untreated syphilis of recent onset. The patient’s kidney ultrasounds were normal, and the relatively small size of her kidneys was consistent with her body size (Figure 1). 

Urinary sediment revealed the presence of poorly preserved “glomerular” erythrocytes, together with small-size cellular casts (Figure 2). 

The patient was treated with antihypertensives and magnesium sulphate for two days, until she developed further seizures in the context of uncontrolled hypertension. Cesarean section was therefore performed, delivering a male baby weighing 1.1 kg (borderline small for gestational age, considering gestational age adjusted for a Ballard score: 6.5 centile), with an Apgar score of 2 at 1 min and 7 at 5 min. 

The baby was hospitalized in the neonatal intensive care unit, where he developed subependimal hemorrhage (Grade I) and severe thrombocytopenia in a context of neonatal syphilis. The baby was hospitalised in the neonatal intensive care unit, where he developed subependimal haemorrhage (Grade I) and severe thrombocytopenia in a context of neonatal syphilis. 

He was treated with support therapy and antibiotics, and was discharged in a good clinical condition four weeks later, having reached a weight of 1.955 kg. 

On account of the patient’s severe proteinuria, not remitting after delivery, and of a clinical pattern that suggested a different kind of glomerular nephritis, the patient underwent a kidney biopsy in puerperium. The kidney biopsy disclosed IgA-dominant postinfectious glomerulonephritis (Figure 3 and Figure 4). The glomeruli presented segments of mesangial proliferation and endocapillary hypercellularity. Edematous endothelial cells occluded some segments and, most importantly, subendothelial electron dense deposits were identified. Interestingly, no sign of glomerular endotheliosis was evident. The immunofluorescence study showed intense IgA positivity with a fine granular pattern with a pseudo-linear distribution, specifically in the capillary loops, without relevant mesangial stain, a pattern which is the hallmark of acute IgA-dominant postinfectious glomerulonephritis.

In the presence of signs of a recent Treponema infection and in the absence of other signs or a history of a different infectious disease, the acute postinfectious glomerulonephritis was considered to be linked to syphilis, and the patient was treated accordingly (benzathine penicillin G 2.4 million units IM per week for 3 weeks). 

At the last follow-up, four months after delivery, both mother and child were well. The mother was in full clinical and laboratory remission, was normotensive and without proteinuria. 

## 3. Discussion

The presence of proteinuria, hypertension and seizures is usually considered to be pathognomonic of preeclampsia (PE), which was in fact the diagnosis made in the first emergency room; however, a formal diagnosis of PE should be based on knowledge of the patient’s history in three different phases. In the first phase, before the 20th gestational week, the patient should have been free of hypertension, proteinuria and other signs of kidney disease; however, in this case, that information was unavailable. The alterations should first appear after the 20th gestational week and should resolve within one or, according to more recent definitions, three months from delivery [1,2,3,17]. Therefore, although a diagnosis of PE is often made, it should be considered as merely putative in the absence of pre-and post-pregnancy data. 

This being said, several elements in the case described supported a diagnosis of PE, including the fact that teen pregnancies and pregnancies in women from disadvantaged backgrounds are at higher risk for PE [20,21,22,23,24]. Low maternal BMI, as in this case case, is an additional minor risk factor [25]. PE is the most common cause of seizures in a pregnant patient without a previous history of epilepsy, however, pregnancy is a predisposing factor for seizures, and hypertensive crises may cause them independently from the presence of a classic PE picture [26,27]. Furthermore, the presence of a small-for-gestational-age baby is common in PE, although it may also occur in the presence of an immunological or chronic kidney disease [28,29,30,31,32].

Conversely, the urinary sediment found in this case was not typical of PE. There are very few studies addressing urinary sediment in PE, and almost all of them are old, and the typical urinary sediment is not clearly defined. While the presence of casts and red blood cells has been described in severe PE, they are probably rare and the urinary sediment is usually bland. For this reason, the presence of an active urinary sediment should be considered a useful tool in the diagnosis of glomerular flares in pregnancy, for example in systemic lupus erythematosus [33,34,35,36,37]. Therefore, in this case, the presence of dysmorphic erythrocytes as well as granular casts was considered highly suggestive of a different kidney disease, probably a proliferative nephropathy (Figure 2) [35,38,39,40]. 

The immunological work-up was negative, but the complement fraction C4 was low, suggesting acute glomerulonephritis, whose classical presentation is usually associated with renal function impairment, rather than severe proteinuria. However, atypical cases are relatively common, especially in adults [41,42]. 

IgA nephropathy, which may present as intense proteinuria in pregnancy, was also a potential explanation, especially in the context of a recent infection [8,43,44]. 

In both cases, a kidney biopsy is needed to guide treatment, and the need for timely diagnosis is more acute in settings, like in Mexico, where the national health system covers expenses (often incompletely) only during pregnancy and the first phase of puerperium [45,46]. 

Our patient was diagnosed with IgA-dominant acute glomerulonephritis (Figure 3 and Figure 4). This emerging nephropathy is sometimes defined as IgA-dominant postinfectious glomerulonephritis, IgA-dominant acute post-streptococcal glomerulonephritis, or IgA-dominant infection-related glomerulonephritis. It is characterized by subepithelial IgA deposits without mesangial involvement, and by a variable clinical course, usually encompassing an acute, nephrotic phase, commonly preponderant with respect to kidney function impairment, as in the case here described [47,48,49,50]. The clinical response is usually favorable when the underlying infection is treated, but in some cases steroid treatment has been added, usually attaining an overall favorable prognosis [47,48,49,50]. 

The disease was first described in 2003 by D’Agati et al., who reported on five patients with type-2 diabetes and acute renal failure occurring after staphylococcal infection [47]. Since then, the disease has been increasingly reported, but remains relatively rare. The largest series published to date encompass 10–20 patients, and the first attempt to pool data, in 2015, gathered less than 100 cases [48,49,50,51,52]. IgA-dominant postinfectious glomerulonephritis has been associated with Staphylococcal infections, but also with *Escherichia coli*, Enterococcus, and other pathogens. For this reason, the generic term “postinfectious” is generally used [48,49,50,51,52]. To the best of our knowledge, no case has so far been associated with Treponema infection.

Conversely, it has been known for decades that syphilis is associated with kidney diseases that are mainly characterized by proteinuria and nephrotic syndrome, and by different immune deposits, including sub-endothelial deposits in which all main immunoglobluins, including IgA, are present [53,54,55,56,57]. The onset of the most commonly reported form, membranous nephropathy, is usually in the early phases of secondary syphilis, often concomitant with a cutaneous rash [58,59,60,61,62]. However, the spectrum of kidney diseases associated with Treponema infection is larger, and encompasses all causes of nephrotic syndrome, namely minimal change disease and focal segmental glomerulosclerosis. Infrequently, it also encompasses proliferative forms, including membrano-proliferative and rapidly progressive glomerunephritis [54,63,64,65,66,67].

In our case, the association with Treponema infection was based on the lack of clinical findings of any other infection coupled with a silent clinical history with regard to other infections during pregnancy, and the patient’s rapid response to anti-Treponema treatment. The role of pregnancy in modulating the clinical picture, with high grade proteinuria and severe hypertension, remains an open question. While a component of superimposed PE cannot be excluded, proteinuria and hypertension remitted only after treatment and did not resolve or improve immediately after delivery, as is usually the case in PE. Furthermore, no sign of endotheliosis, the morphologic hallmark of PE, was found, thus supporting the hypothesis of a concomitant glomerulonephritis more than that of a mixed picture (Figure 3 and Figure 4).

## 4. Conclusions

This case suggests that IgA-dominant postinfectious nephritis should be added to the spectrum of syphilis-associated glomerulonephritis and underlines the need for a careful differential diagnosis in all cases of presumed PE. This could be lifesaving, especially in disadvantaged populations, where pregnancy represents the first and often only occasion in which a young woman goes through a medical check-up. While complex analyses, including kidney biopsy, may be needed for diagnosis, simple and inexpensive tests, including urinary sediment analysis, are important diagnostic tools that can suggest the presence of kidney diseases other than PE. 

## Figures and Tables

**Figure 1 jcm-08-00114-f001:**
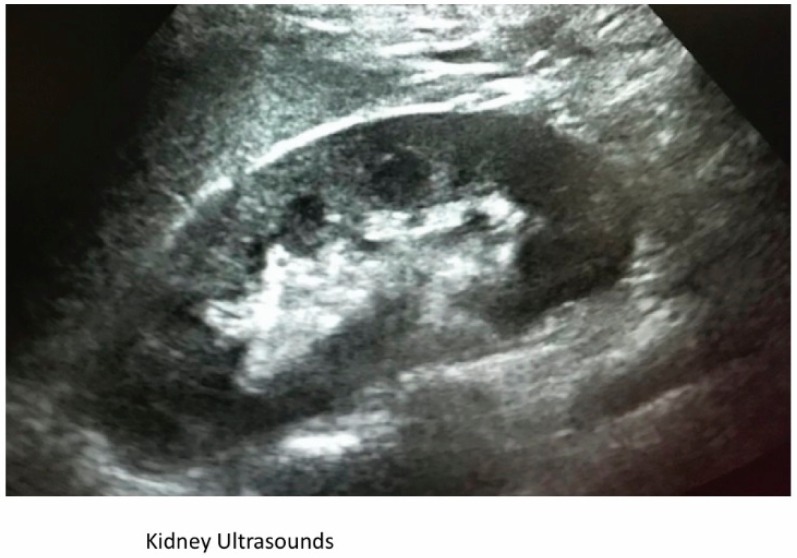
Ultrasound showing a kidney of normal size and echogenicity.

**Figure 2 jcm-08-00114-f002:**
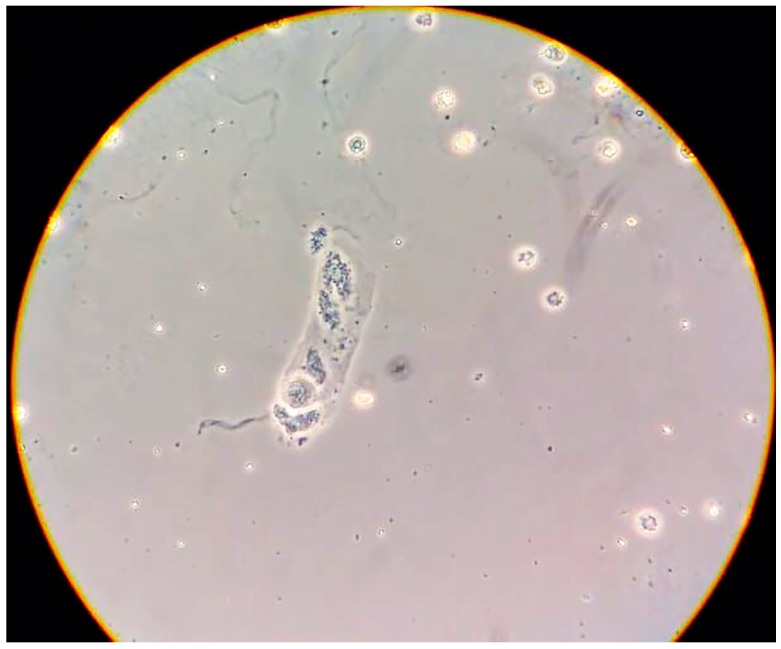
Urinary sediment with dysmorphic erythrocytes and a small granular cast.

**Figure 3 jcm-08-00114-f003:**
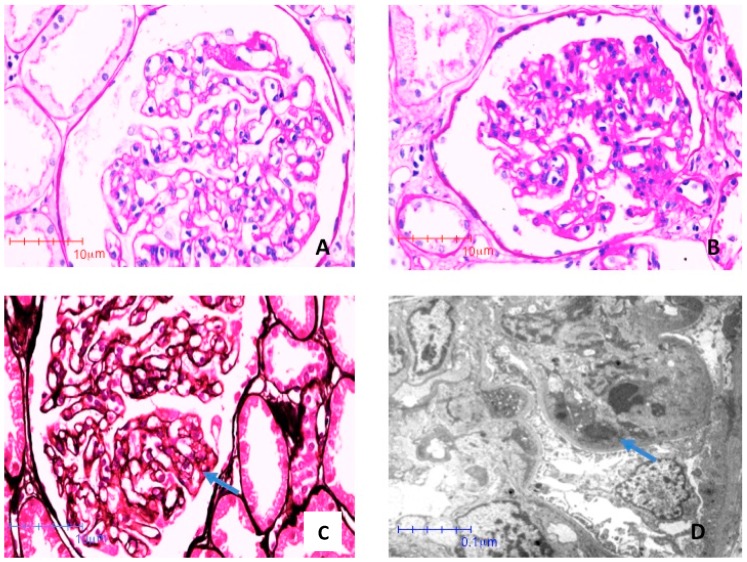
Photomicrographies stained with Periodic Acid Shiff (PAS) (**A**) and (**B**) and Jones Metenamine (**C**), respectively (40×). The glomeruli showed segments of mesangial proliferation and endocapillar hypercellularity. Edematous endothelial cells occlude some segments (arrow). (**D**) Electron photomicroscopy (8000×). Subendothelial electron dense deposits (immune complexes) are indicated by arrows.

**Figure 4 jcm-08-00114-f004:**
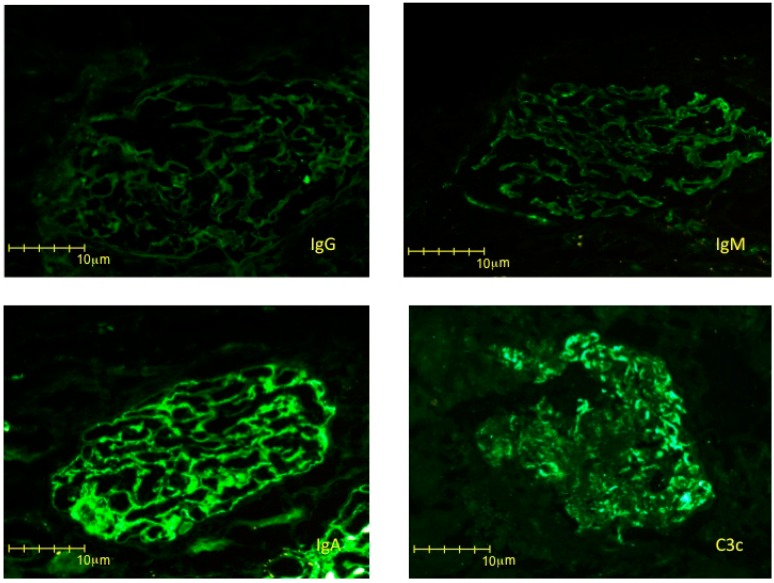
Direct immunofluorescence study with four immunoreactants. IgG and IgM, were negative. IgA showed positivity with a fine granular pattern, specifically in the capillary loops (3+). Mesangial stain is minor. C3c showed granular deposits, especially in the capillary walls (2+).

**Table 1 jcm-08-00114-t001:** Summary of the main biochemical tests.

Parameter	Initial Testing	1 Week after Delivery (Kidney Biopsy)	4 Months after Delivery
Albumin, g/dL	2.3	-	4.7
Creatinine, mg/dL	0.8	0.8	0.7
Urea, mg/dL	43	32	26
Hemoglobin, g/dL	9.1	9.9	14.2
LDH UI/L	615	677	414
AST, UI/L	30	33	18
ALT, UI/L	40	46	27
Glucose mg/dL	73	71	72
Uric acid mg/dL	7.9	8	5.1
Proteinuria/creatinuria g/g	6.42	4.9	0.036
Erythrocytes n /ml	341	-	2
VDRL	1:128	-	1:32
Syphilis IgM	Positive	-	-
Syphilis IgG	Positive	-	-

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
