# Peer review of "Acute IgA-Dominant Glomerulonephritis Associated with Syphilis Infection in a Pregnant Teenager: A New Disease Association"

_jcm, 2019, doi:10.3390/jcm8010114_

Round 1
Reviewer 1 Report
The authors presented in this manuscript a case report of a young woman exhibiting syphillis associated to IgA postinfectious glomerulonephritis during her pregnancy. The case is well described and the conclusions will be helpfull for clinicians.
Did the authors detect circulating Syphillis IgA or IgG during the infection?
There are two missing citations in the discussion showing renal IgA deposits with other Igs after syphillis infection:
Tourville DR et al. Am J Pathol 1976
Schillinger F et al. Presse Med 1983
Author Response
Thank you for your kind comments and suggestions;
Did the authors detect circulating Syphillis IgA or IgG during the infection?
No, the test for IgA was not available.
There are two missing citations in the discussion showing renal IgA deposits with other Igs after syphillis infection:
Tourville DR et al. Am J Pathol 1976
Schillinger F et al. Presse Med 1983
thank you very much, we added these pivotal papers in the reference list and in the discussion.
thanks for your kind comments and suggestions,
the authors
Reviewer 2 Report
This is a strong case report, incorporating an unusual presentation of post-infectious glomerulonephritis, newer understandings of IgA related glomerulopathy, and awareness of the importance of syphilis as a cause of unusual glomerular disease; and this is all within a setting where a clinician might be looking elsewhere, i.e., diverted by the more apparent diagnosis of pre-eclampsia; thus highlighting the importance of avoiding anchoring bias, and using clinical tools such as urinalysis and biopsy to their greatest advantage. There are minor instances where the English is slightly awkward. But overall, this is a very worthwhile case report.
Author Response
thank you very much for your kind words.
none of us is mother language, and we had the paper reviewed by Susan Finnel, who is an American lecturer in our University. However, she's not a physician and some awkward expressions may escape.
MDPI makes a last editing, that should smooth the language...